# Therapeutic Approaches in Pancreatic Cancer: Recent Updates

**DOI:** 10.3390/biomedicines11061611

**Published:** 2023-06-01

**Authors:** Lokender Kumar, Sanjay Kumar, Kumar Sandeep, Sanjay Kumar Singh Patel

**Affiliations:** 1School of Biotechnology, Faculty of Applied Sciences and Biotechnology, Shoolini University, Solan 173229, India; lokenderkumar@shooliniuniversity.com; 2Department of Life Sciences, School of Basic Sciences and Research, Sharda University, Greater Noida 201310, India; sanjay.kumar7@sharda.ac.in; 3Dr. B.R.A. Institute Rotary Cancer Hospital, All India Institute of Medical Sciences, New Delhi 110029, India; drkumarsandeep@aiims.edu; 4Department of Chemical Engineering, Konkuk University, Seoul 05029, Republic of Korea

**Keywords:** molecular mechanism, cancer, biomolecules, therapeutics, pancreas, health

## Abstract

Cancer is a significant challenge for effective treatment due to its complex mechanism, different progressing stages, and lack of adequate procedures for screening and identification. Pancreatic cancer is typically identified in its advanced progression phase with a low survival of ~5 years. Among cancers, pancreatic cancer is also considered a high mortality-causing casualty over other accidental or disease-based mortality, and it is ranked seventh among all mortality-associated cancers globally. Henceforth, developing diagnostic procedures for its early detection, understanding pancreatic cancer-linked mechanisms, and various therapeutic strategies are crucial. This review describes the recent development in pancreatic cancer progression, mechanisms, and therapeutic approaches, including molecular techniques and biomedicines for effectively treating cancer.

## 1. Introduction

Pancreatic cancer is a fatal disease that affects the pancreas, a large gland in the abdomen. American Cancer Society suggested that pancreatic cancer is the third highly common type of cancer in humans. Initial recognition is critical so that patients receive the extremely efficient treatment possible and have the greatest chance of survival [1,2,3]. Primarily, many patients are detected at an advanced stage. In the United States, this year approximately 53,090 people will be diagnosed with pancreatic cancer, and more than 41,170 people died from the disease [4,5]. Therefore, we must continue identifying new and improved ways to diagnose the disease earlier to save lives. Pancreatic cancer is highly destructive and has a poor prognosis with only a 9% of survival rate within five-years. Unluckily, the symptoms of pancreatic cancer are often vague and easy to overlook, so many people do not know that they have it until it is too late [6,7]. Its primary symptoms include jaundice, loss of appetite/weight, and right abdomen pain [4,8]. The pancreas is also challenging to reach and diagnose, so traditional imaging methods often cannot detect cancer early enough. However, early diagnosis and treatment are critical to improving survival. Today’s early detection methods are better than ever, but there are still significant limitations to available diagnostic tools. Methods of detecting pancreatic tumors include x-rays, ultrasounds, MRI.s, and CT scans [4]. These tests can effectively identify the tumor’s location and the extent of the disease. It is difficult to diagnose early, but specific blood tests can detect it if certain risk factors exist. Specific biomarkers can also be applied to detect cancer cells [9,10]. Pancreatic cancer is diagnosed with endoscopy, a minimally invasive procedure that uses a tube with a tiny camera attached to examine the inner lining of the digestive tract. During the process, removing a tissue sample from the pancreas to test for cancer cells can be validated [11,12]. Further, staging determines how far your cancer has spread and whether it has spread beyond the pancreas and risk assessment to effective diagnostic management [13]. 

The use of drugs destroys cancerous cells while sparing the surrounding healthy cells, allowing them to be combined with other cancer treatments to maximize their effectiveness [14,15]. Existing treatment possibilities for patients with advanced pancreatic cancer include chemotherapy, immunotherapy, targeted therapies, and surgery [4,16,17]. Chemotherapy involves using powerful drugs that kill cancer cells by attacking their DNA and disrupting their growth and division [18]. Gemcitabine is the first drug approved to treat pancreatic cancer [19]. It stops the development of new cancer cells and accelerates the immune system to kill them. The current standard treatment of gemcitabine is an intravenous injection twice a day for several days. After that, the treatment is repeated every three weeks until the patient no longer responds to the drug or until their condition worsens [19,20]. Unfortunately, few patients with advanced-stage cancer will respond to these treatments. New targeted therapies continue to be consideration-based in genomics, genetics, and molecular therapies, along with integrative concepts for improved diagnostics and treatments of pancreatic cancer [21,22,23].

Commonly known pancreatic tumors are pancreatic ductal adenocarcinoma (PDAC), accounting of nearly 90% of pancreatic-based cancers [24,25]. In some cases, patients may also develop secondary tumors in the liver or lungs due to the disease spreading throughout the body [26]. In general, treatment options for pancreatic cancer are usually limited and ineffective. Many patients with this disease do not respond to traditional treatments and are only given a few months to live [4,27]. However, recent developments in cancer treatment have led to many new therapies that may extend the lives of patients diagnosed with pancreatic cancer [28,29]. Furthermore, global perspectives have been significantly altered in the current pandemic toward therapeutic strategies for sustainable approaches to maintain good health and the economy [30,31].

Although plenty of excellent reviews are available on pancreatic cancer [32,33,34,35,36,37], this review stands out as it offers an in-depth analysis of the most recent studies and advancements in the field. It thoroughly examines the current understanding of the progression of this disease and explores a wide range of therapeutic strategies that show potential in treating pancreatic cancer. These strategies include traditional chemotherapy, surgical interventions, and cutting-edge approaches such as immunotherapy and targeted therapies. This review evaluates these treatment option’s effectiveness, limitations, and possible side effects. It aims to provide a comprehensive perspective on the advancements made in pancreatic cancer treatment.

## 2. Pancreatic Cancer Molecular Manifestation and Pathways Regulation

Pancreatic cancer is challenging, with a poor progression and limited treatment options [38]. Its tumorigenesis or metastasis involves, pathways, including phosphoinositide 3-kinase (PI3K)/protein kinase B (AKT), RAS, janus kinase (JAK)/signal transducer, and activator of transcription (STAT), NF-κB, Hippo/yes-kinase-associated protein (hippo/YAP), and Wingless/int1 (WNT). These pathways are associated with numerous cellular processes linked to pancreatic cancer, such as apoptosis, angiogenesis, differentiation, immunological regulations, metabolism, migration, and cell proliferation. In addition, histone modification is a vital feature in pancreatic cancer for epithelial-to-mesenchyme transition [4,39]. The regulation and consideration of these pathways for pancreatic cancer can be helpful in developing novel targets and therapeutics.

A key feature of pancreatic cancer includes the immunosuppressive tumor microenvironment [40,41]. Several molecular and cellular factors have been identified as critical players in the induction and maintenance of immunosuppression within the pancreatic stroma. The pancreatic stroma is composed of an extracellular matrix, immune cells, and fibroblasts that surround the tumor cells, forming a barrier that impedes cancer drug effect and immune cell infiltration [42] (Figure 1). Macrophages, which are a critical type of innate immune cells, play a significant role in immunomodulation in pancreatic cancer via a secreting range of cytokines [43]. Cytokines play a crucial role in tumor growth and immune cell evasion by promoting cancer cell invasion, proliferation, and immunosuppression [44,45]. Inflammatory cytokines such as IL-1β, IL-6, IL-8, and IL-10, have been demonstrated to activate tumor-associated macrophages (TAMs). The cytokines and chemokines attract immune cells, i.e., regulatory T cells (Treg cells), TAMs, and neutrophils, which impede CD8+ cytotoxic T cells function. Regulatory T cells (Tregs) have been associated with advancing pancreatic cancer by curbing cytotoxic T cells [46]. Pancreatic cancer cells coordinate immune evasion by synchronizing the secretion of cytokines in a highly coordinated way through TP53-dependent or KRAS-dependent pathways [47]. Tregs are vital in preserving immunologic self-tolerance and regulating suppression in pancreatic tumor growth.

Recent studies have emphasized the vital importance of the cancer cell microenvironment in advancing cancer progression [48]. Moreover, the immune cells within the cancer cell microenvironment, including B cells, natural killer cells, T cells, and myeloid-derived suppressor cells are dysregulated and fail to mount an effective antitumor response [49]. The immunosuppressive cancer micro-environment shows a decisive role in the advancement of therapeutic resistance, which explains why current treatments are ineffective against pancreatic cancer. The progression of pancreatic cancer is modulated by various signaling pathways, including NF-κB, JAK/STAT, PI3K/AKT, Hippo/YAP, RAS, and WNT pathways (Figure 2). These pathways impact cellular functions such as apoptosis, differentiation, immunological regulation, metabolism, migration, angiogenesis, and cell proliferation [50,51,52,53]. Exploring and managing these signaling pathways has the potential to uncover novel targets and therapies for pancreatic cancer. RAS is a crucial driver of effector pathways, and its oncogenic activation is frequently found in pancreatic cancer, particularly in the KRAS isoform [47]. This activation can lead to cell proliferation, transformation, metastasis, and pro-inflammatory signaling activation. The PI3K/AKT pathway plays a significant role in pancreatic cancer regulation, with great potential for therapeutic targeting [54]. The PI3K/AKT pathway is commonly triggered in pancreatic cancer, with abnormal AKT overexpression associated with poor prognosis. PI3K signaling controls pancreatic cell plasticity and is activated early in tumor evolution [55]. This pathway is also activated by insulin-like growth factors and abnormal expression of various noncoding RNAs [56]. Several PI3K/AKT inhibitors are being investigated for their potential therapeutic effects in pancreatic cancer patients. 

NF-κB is a crucial transcription-factor that shows a significant role in inflammation, is frequently stimulated in pancreatic cancer, and promotes cancer development, metastasis, and drug-resistance [57]. KRAS and other oncogenic mutations can also activate NF-κB. High quantities of chemokines are seen in pancreatic cancer, forming a positive feedback loop that enhances NF-κB signaling. Several ncRNAs can regulate the NF-κB pathway [58]. NF-κB is involved in antitumor immunity, and inhibitors of the pathway show promise as a therapeutic option. JAK/STAT pathways are engaged in a range of human cancers, together with pancreatic cancer, and increased JAK2 signaling indicates an inadequate prognosis of the disease [58,59]. The signaling pathway is implicated in inflammation in pancreatic cancer. In addition, interferons can increase PD-L1 expression via directly/indirectly influencing JAK-STAT signaling. Sustained JAK-STAT initiation can advance chronic inflammation and may impede CTL activation. JAK-STAT pathways have been associated with cancer developmental progress. YAP-TAZ are the major contributing factors of the Hippo pathway in pancreatic cancer [50]. Studies show that YAP is extremely articulated in patients and linked with an inadequate diagnosis of the disease. YAP is needed for cancer progression and is a crucial player in KRAS mutant mice. YAP can lead to disease relapse in the deficiency of KRAS and plays a vital driver of squamous pancreatic cancer. YAP-TAZ are transcriptional co-activators that can drive the gene expression engaged in proliferation and cell survival, impacting the hostile behavior of pancreatic cancerous cells. YAP also regulates the immunosuppressive microenvironment by modulating the behavior of PSCs and influencing the enlistment of TAMs and MDSCs. The WNT pathway controls somatic stem cells and is involved in pancreatic carcinogenesis and tumor progression through canonical and noncanonical pathways [60,61]. KRAS activation promotes pancreatic cancer cell movement and infiltration through the WNT pathway, and increased WNT/β-catenin signaling enhances the stem cell-like phenotypes. Both canonical and noncanonical WNT ligands have been linked to pancreatic cancer progression [60]. 

## 3. Therapeutic Strategies

### 3.1. Non-Coding RNAs

Non-coding RNAs (ncRNAs) are essential in numerous progressions, including cancer, heart, and neurological diseases [4,62]. Micro-RNAs (miRNAs) or synthetic antagomirs are ncRNAs composed of ~22 long nucleotides. These are proposed to be associated with regulatory functions, i.e., cell proliferation, apoptosis, and autophagy. In addition, studies have shown that miRNAs such as miR-203 possess antitumor potential by regulating gene expressions [36]. However, miRNAs released in the bloodstream can be used as markers or indicators to monitor pancreatic cancer progression or aggressiveness. The role of a few miRNAs in cancer-transforming cell metabolism includes as follows: (i) miR17-92 and miR-21 in proliferation inhibition, (ii) miR-126 in anti-oncogene action inhibition, (iii) miR-15b, and miR-155 in mutations accumulation, (iv) let-7d, miR-23b, miR-126, and miR-200c in inflammation promotion, (v) miR-10b and miR-29 in metastasis activation, (vi) miR-21 and miR17-92 in the immune cells elimination inhibition, and (vii) let-7, miR-16, miR-21, and miR-221/222 in replication immortality [36]. miR-203 helps suppress cell invasion and migration by caveolin-1 and is down-regulated in pancreatic cancers [4]. Various miRNAs involved in pancreatic cancer are miR-21, miR-155, miR-221, miR-222, miR-376a, and miR-301 [36]. In addition, it can hinder the tumorigenic properties of pancreatic cancerous cells by regulating DJ-1 expression, and PTEN-PI3K/AKT pathway. In contrast, over-expressed miR-203 influences pancreatic cell proliferation and apoptosis by alteration of SOCS3 expression [62]. Although, limited studies have been demonstrated on the mechanism and critical role of miR-203 in pancreatic cancer, which requires more analysis for future studies [36,62].

### 3.2. Cyclin-Dependent Kinases

Protein kinases work an essential role in protein function alteration by phosphorylation to enhance or decline biological activities, i.e., transcription and translation [4,24]. However, protein kinase activity regulation is key in therapeutics against various diseases, including autoimmune, neural, and cardiovascular. Cyclin-dependent kinases (CDKs) such as CDK1-CDK4 and CDK6 serine/threonine kinases are primarily involved in cell cycle regulation by the mechanism of phosphorylation, interaction between proteins, or ubiquitin-based proteolysis. CDKs activation requires its binding to the cyclin subunit. To initiate cells from G2 to the mitosis phase, CDK1 complexes with cyclins A and B. On the other hand, CDK2 links to cyclins A and E, which is essential in the G1-S transition and in the S phase [24]. Alteration in cell cycle regulation is an essential phase towards the transition of normal cells to cancerous cells primarily associated with the functional de-regulation of CDKs. CDKs have been considered potential targets in cancer therapeutics. A few CDKs inhibitions are known, such as abemaciclib, dinaciclib, Palbociclib, ribociclib, and RO-3306, and are helpful in the treatment of cancer at metastatic or advanced cancer stage [24]. 4-benzyl-1-piperazinyl-phthalazine compounds (8 g, 10 d and 10 h) are exhibiting anti-proliferation activity towards MDA-PATC53 with 50% of viability inhibitory concentration (IC_50_) of 0.51–0.88 μM and PL45 with IC_50_ of 0.74–1.14 μM). These phthalazines highly showed inhibitory selectivity against CDK1 with IC_50_ of 36.8–44.5 nM. In contrast, a minor inhibitory influence of these compounds was noted against other kinases, including CDK2, CDK5, IGF1R, AXL, BRAF, FGFR, JAK1, and PTK2B [24]. 1-Piperazinylphthalazines (16 k, and 21 d) impeded VEGFR-2 with IC_50_ of 0.35–0.40 μM [63]. In mechanism, cell cycle controlling mechanisms regulates the growth of cells. Inhibiting cell proliferation can be helpful to cancerous cell therapy targets by cell cycle arresting at G2/M check-points [24,63]. For cancer-treating drugs, the cell cycle arrest mechanism initiation is anticipated via their influence on cell toxicity. Phosphorylation of E2F1 and CDC6 by CDK2/cyclin A involves S phase termination and pushes the S phase to the G2 phase transition. Furthermore, CDK1 activation by cyclin A leads to cells being put in the M phase. During mitosis, the CDK1/cyclin B complex maintains CDK1 activity, which is vital for the progression of the cell cycle. Therefore, the alteration in CDK1 expression assists in separating chromosomes, mitosis accomplishment, and cytokinesis during cellular replication [24,63].

### 3.3. Oncolytic Virus

PDAC escape the antitumor immune-system through inadequate angiogenesis, massive-stroma, and immune-suppressive cells infiltration [7]. Consequently, a unique therapeutic strategy to accelerate the immune response against antitumor is considered necessary to enhance the efficacy of immunotherapy against PDAC. Inhibitors of immune checkpoints, such as antibodies towards anti-programmed cell death 1 (PD-1), are proven effective in cancer treatment [4,7]. In PDAC, the deprived immune response of PD-1 obstruction therapy is refractory. Therefore, novel oncolytic virotherapy is a potential therapeutic method via immunogenic cell death [64]. Oncolytic adenovirus OBP-702-mediate over p53 expression that can stimulate immunogenic cell death via antitumor immune responses towards human PDAC by releasing damage-associated molecular patterns, i.e., extracellular delivery of high mobility-group box 1 protein, and adenosine-triphosphate. A subcutaneous PAN02 tumor (PDAC) model suggested that OBP-702 involves tumor permeation of T cells (CD8^+^), and PD-1 blockade antitumor efficacy [64].

Mutation in a rat sarcoma virus (RAS) is associated with nearly 30% of the entire human cancers and accounts for almost 85% of Kirsten RAS (KRAS) of whole RAS mutations [65]. Pancreatic cancers are linked to the KRAS gene’s oncogenic mutations in approximately 90% of cases. KRAS is a component of epidermal growth factor receptors (EGFR) signaling that are vital molecules in cancer therapy. In a mechanism, KRAS triggers the downstream growth-associated signaling pathways and quickly enhances phosphatidylinositol 3-kinase (PI3K)/AKT, and mitogen-activated protein kinase (MAPK). Cetuximab is used as a therapeutic towards EGFR in case of metastatic colorectal cancer individual who does not respond to chemotherapy. The application of cetuximab is restricted to wild-type KRAS. However, numerous attempts have been initiated to create molecules that act against tumors by inhibiting the activity of mutant KRAS [65]. A decline in specific miRNA’s intracellular contents can provoke genetic information transmission that leads to disease progression. For example, compared to normal cells, the MIR143 expression that p53 transcribes declined by nearly 80% in colorectal tumors and PDAC. The mechanism of miRNA therapy involves rebuilding tumor cells to normal by substituting down-regulated miRNA or their suppression [65]. Chemically altered MIR143 considerably inhibits tumor growth by affecting AKT, KRAS, extracellular signal-regulated kinases (ERK), and son of sevenless homolog 1 (SOS1). In addition, chemically modified MIR143 (MIR143#12) anticancer effects involve inhibition of KRAS networks in pancreatic cancer cells with/without KRAS mutation and exhibit high-level RNA nucleases resistance activity [4,65]. miR-143 #12 showed superior anticancer activity (IC_50_ = 0.36 nM) to siR-AKT, siR-KRAS, or siR-ERK2 silencing (IC_50_ > 20 nM) toward BxPC-3 and MIA PaCa-2 cells. In contrast, PANC-1 exhibits higher IC50 of 4.75 nM for MIR143 than up to 0.98 nM for siR-AKT, siR-KRAS, or siR-ERK2 at the incubation of 96 h [65]. Here, MIR143 acts by cell cycle arrest autophagy or apoptosis. Overall, the cell proliferation inhibition was observed quite similarly for siR-KRAS and MIR143 towards pancreatic cancer KRAS mutant cells. Further, the effectiveness of MIR143#12 towards PANC-1 can be potentially improved by its supplementation with siR-KRAS/cetuximab. However, MIR143#12 showed negligible adverse influence on normal cell growth; thus, developing a suitable drug delivery system for its delivery towards specific tumors is essential for side effects assessment [65].

### 3.4. Nanomedicine

Nanomedicine is a promising approach that can be potentially applied for treatment, including cancer treatment. Still, the clinical use of nanomedicines is limited by various factors—(i) low loading of drugs, (ii) tumor target specificity, (iii) potential cytotoxicity, and (iv) desirable mass production. Regarding industrial and technical attributes, the alternative strategy for developing carrier-free prodrug nanoparticles seems more promising. A conjugate of doxorubicin (DOX) and cathepsin B specific cleavable peptide (FRRG) formulated via self-assembly followed Pluronic F68 stabilization as F68-FDOX prodrug nanoparticles exhibiting high loading of >50% drug [66]. F68-FDOX proficiently gathers within tumors via enhanced permeability and retention influence with further release of DOX via cathepsin-B specific cleavage process exhibited significant antitumor activity for MDA-MB231, HT29, and KPC960. In vivo analysis suggested that F68-FDOX is highly safe as no toxicity was noted toward normal cells [66]. Dextran-coated maghemite nanoparticles (56 μg/mL) exposure reduced ~50% of PANC-1 cell viability after 72 h incubation via alteration in expressions of heat shock proteins (HSPs) and p53 protein [67]. Lipid nanoparticles (LNP_DTX-P_) on gold nanoparticles with docetaxel prodrug showed enhancement in the uptake of nanoparticles up to 2.8-folds by MIA PACA-2 cells than the controls [68]. In vitro assay, MIA PACA-2 cells proliferation IC_50_ of 9.8 nM for LNP_DTX_ noted against only docetaxel prodrug (44.4 nM). In mechanism, nanoparticles containing cancerous cells cannot divide and are trapped in the M phase. Hence, LNP_DTX-P_ on gold nanoparticles can be potentially employed for radiotherapy-based pancreatic cancer treatments [68]. The combined approach of using magnetic nanoparticles and hyperthermia against BxPC3 pancreatic tumor cells proved beneficial over nanoparticles only [69]. Nanoparticles enhanced DNA double-strand breaks via radio sensitization effects and ROS production. Here, a combined treatment strategy exhibited significantly higher BxPC3 cell death after 6 h of incubation than irradiation or nanoparticle administration [69]. MIA-PaCa-2 treated by 4 μM of polyethyleneglycol-betulinic acid (PEG-BA) polymer-drug conjugate exhibits enhancement in expression of proapoptotic genes *TNF* and *CASPASE 3* up to 23.7-, and 12,060-fold, respectively [70]. BA helped in anti-inflammatory and antioxidant activities in this conjugate, and the conjugate showed IC_50_ of 15.6 μM compared to the antioxidant potential for BA-only of >100 μM. These conjugates induce apoptosis-mediated death of MIA-PaCa-2 by arresting the sub-G1phase of cell cycles [70].

### 3.5. Adjuvants, Immunological Targets and Peptide

In Asia, S-1 adjuvant-based chemotherapy is primarily adapted to treat PDAC and its dose intensity highly influences survival potential [71]. Pancreatic cancer patients with S-1 dose intensity of ≥62.5% proved beneficial in survival with a median period of 53.3 months compared to its lower dose with a survival period of 20.2 months. This finding suggests that S-1 adjuvant chemotherapy is an effective therapeutic against pancreatic cancer for prolonged postoperative survival by maintaining a dose intensity above 60% [71].

Immune check-points inhibitors like antibodies PD-1/PD-L1 and CTLA-4 are applied for immunotherapy against cancers [64]. However, PDAC is not as sensitive for immunotherapy because of cytotoxic T cell inferior infiltration and limiting mutation burden. Immunological therapy that is associated with molecular compounds acting towards toll-like receptors (TLRs) can be a robust approach for treating cancers [19]. Erlotinib, an immunomodulatory drug linked with gemcitabine, works by impeding an epidermal-growth factor receptors tyrosine kinase. Compounds like protein aggregate magnesium-ammonium phospholinoleate-palmitoleate anhydride (P-MAPA) are emerging potential candidates against pancreatic cancer involving the TLR4 signaling pathway [19]. P-MAPA and gemcitabine-coupled therapy efficiently reduced the progression of neoplastic lesions and histopathological improvement by up to 80% in pancreatic cancer over only 40% repair using P-MAP alone. P-MAPA and P-MAPA/gemcitabine therapies enhance TLR4 protein contents that improve interferon signaling pathways via abnormal cell proliferation suppression [19].

In PDAC, thrombospondin-1 (TSP-1) associated with cancer-associated fibroblasts (CAFs) induces downregulation of Smad4 expression and enhances malignant potential via activation transforming growth factor-β (TGF-β) signals [72]. Smad4 involves TGF-β superfamily cytokines signal transduction by target genes transcriptional activation and plays a vital role in pancreatic tumor suppressor. Deletion of the Smad4 gene (*DPC4*) activates faster cell invasiveness and metastatic capability. In vivo, TSP-1) acts as an essential activator of TGF-β. In PDAC cells, *DPC4* knockdown triggered TSP-1 overexpression along with TGF-β signal initiation. Also, the overexpression of TSP-1 caused Smad4 expression downregulation and improved the proliferation of cells. LSKL peptide (TSP-1 inhibitor) treatment prevents TSP-1 binding to a latency-associated peptide (LAP) and TGF-β signal activation that results in anticancer impacts via attenuation of cell proliferation, migration, and chemoresistance in PDAC cells. However, LSKL peptides as TSP-1 inhibitors can potentially be therapeutics in PDAC patients [72].

### 3.6. Natural Bioactive and Organic Compounds

Natural-derived biomolecules are proven to be beneficial for broad biotechnological applications [73,74,75]. Curcumin has been shown to have a broad target and involve in the regulation of many cellular signaling pathways such as NF-κB/COX2, WNT/β-catenin, SHH, STAT3, NOTCH, TGF/Smad, and PI3K/AKT. These are participating in the development and progression of various cancers [74,75,76,77,78,79,80]. Curcumin induces apoptosis, inhibition of progression, and angiogenesis in pancreatic cancer. For curcumin, IC_50_ of 8 µM, 20 µM, and 12 µg/mL against BxPC3, and Panc1 has been noted, respectively [74,80,81]. It has been reported that curcumin down-regulates NF-κB binding, and the activity of Ikappa B kinase, which leads to decreased cell proliferation and enhanced apoptosis in pancreatic cancer [78]. Another study further supported this, where the author reported curcumin treatment with a pancreatic cancer cell line, improving the upregulation of FOXO1 expression [79,80]. Additionally, Curcumin has been displayed to pancreatic cancer cells to sensitization to gemcitabine drug, which failed to treat pancreatic cancer cells due to drug resistance [76,81]. Hence, curcumin was considered potential therapeutics for inhibiting the growth of pancreatic cancer stem cells.

Chaetospirolactone (CSL) from *Chaetomium* spp. has been described to increase the susceptibility of drug-resistant cells of pancreatic cancer to make it prepare for apoptosis via epigenetic regulation [82]. Additionally, it enhances death receptor 4 (DR4) expression. Further, 100 nM of CSL treatment for 24 h reverses TRAIL resistance in PANC-1 through DR4 epigenetic-regulation. CSL treatment suppresses zeste homolog 2 (EZH2) and lower histone H3 lysine 27 trimethylation (H3K27me3) for transcription of DR4. Altogether, its administration might be a presumed approach against pancreatic cancer [82,83]. The treatment by cordifoliketones A from *Codonopsis cordifolioidea* inhibits growth, and induces apoptosis of PANC-1, AsPC-1, and BxPC-3 [84]. In vitro treatment of these cells by 2–6 µg/mL of cordifoliketones A up to 48 h showed apoptotic influence without influencing normal cells. At IC_50_ of 4.2 µg/mL cordifoliketones A comparative treatment to BALB/c nude mice holding PANC-1, AsPC-1, and BxPC-3 alone (placebo) and PDAC-cells showed that PDAC exhibits lover proliferation than placebo [83,84]. The treatment by 100 µg/mL of Danggui-Sayuk-Ga-Osuyu-Saenggang-Tang (DSGOST) as a Korean traditional medicine inhibited vascular endothelial growth factor (VEGF) [85]. Dicatenarin, a secondary metabolite isolated from *Allium schoenoprasum* has been reported to have inhibition properties against the NCI60 pancreatic cancer cells with a IC_50_ of 12 µg/mL [86]. Furthermore, it enhances the ROS-mediated mitochondrial permeability transition, leading to the upregulation of caspase-3 apoptotic protein in MIA PaCa-2 pancreatic cell [87]. In addition, a IC_50_ of 12 µg/mL were determined against MIA PaCa-2 cell line [86,87].

Diphyllin is recognized as promising natural therapeutics against pancreatic cancer [88]. Still, its application in pancreatic cancer is limited due to partial solubility, inadequate potency, and weak metabolic stability. Therefore, the derivatization diphyllin can be beneficial to overcome these limitations. The nitrogen-based derivatization of diphyllin results in superior metabolic stability and aqueous solubility compared to pure diphyllin. The amino-derivatized compound 15 showed IC_50_ of 3 nM against CFPAC-1 cells with 69-fold higher effectiveness over diphyllin as control [88]. These amino derivative compounds dose-dependently stimulated cell cycle arrest (G0/G1 phase) via CDK4 and cyclinD1 down-regulation. In the xenograft model, a compound 15 dose of 10 mg/kg treatment showed high potent efficacy against transplanted PANC02 tumors via growth suppression with no noticeable safety anxiety [88]. Diosgenin, a derivative of *Solanum*, *Dioscorea*, and *Costus* spp., is demonstrated to stimulate apoptosis and cell cycle arrest in pancreatic cancer cells PANC-1 and Patu8988. Furthermore, diosgenin downregulates an expression of oncogenic protein EZH2 and its target vimentin and upregulates PTEN [89,90]. Therefore, it could be a promising therapeutic compound for PC treatment targeting EZH2. In addition, 75 µg/mL diosgenin for 72 h has been reported to suppress 70% growth of Patu8988 and Panc-1 cell lies [83,89,90].

Echinacoside (ECH), a derivative of the plant *Cistanchessalsa*, has been reported to enhance apoptosis via increasing ROS and lowering MMP in SW1990 [91]. Since the higher level of ROS and reduced MMP expression have been associated with apoptosis induction. Furthermore, ECH upregulates the expression of Bax via tumor suppressor p53. Also, it enhances apoptosis through the MAPK pathway, leads to reduced JNK, and ERK1/2 activities, and enhances p38 activity. Additionally, it is not involved in the AKT pathway, which is required for cell proliferation [83]. Furthermore, maximum growth inhibition was observed at 72 h using 100 µM of Echinacoside for SW1990 cell lines [91]. Elemene, a plant *Zingiberaceae* extract inhibits cells proliferation, and enhanced cell cycle arrests in cells of BxPC-3 and PANC-1 [92]. Furthermore, its administration is associated with apoptosis induction via upregulating p53 and downregulating Bcl-2 in BxPC-3 carrying BALB/c nude mice-model [83,93]. 60 µg/mL of elemene has been observed to induce more than 85% of cell death after 72 h in BxPC-3 and Panc-1 cells [93]. Methyl4-(2-isothiocyanatoethyl)benzoate (isothiocyanates) a key cruciferous vegetables compound is known for preventing cancer properties. Luo et al. [94], demonstrated that isothiocyanates had a more apoptotic impact PANC-1 than control cells. It exhibits up-regulation of ROS, and GSH down-regulation that results in cancer cells apoptosis [83,94]. Furthermore, optimal apoptosis was observed at dose 10 µM Methyl4-(2-isothiocyanatoethyl)benzoate after 72 h treatment in Panc1 and Capan2 cell lines [94,95]. A combined treatment of monogalactosyl diacylglycerol (MGDG), and radiation exhibited higher apoptosis than controls e for inhibiting growth of tumor in xenografts mouse [96]. The cell death was observed at IC_50_ of 25.6 µM againsts cancer cells (AsPC-1, BxPC-3, PANC-1, and MIAPaCa-2) at 72 h of incubation. Wang et al. [97], described piperlongumine from *Piper longum*, up-regulates c-PARP, and procaspase-3 in various pancreatic cancer cells (AsPC-1, BxPC-3, and PANC-1). Tum growth was significantly inhibited in BxPC-3 carrying BALB/c mice by piperlongumine (10 mg/kg) over 21-days. In vivo and in vitro, the apoptotic effects of piperlongumine were enhanced by supplementation of gemcitabine. Additionally, piperlongumine at dose 40 µmol/L inhibit maximum cell growth in PANC-1 after 72 h treatment [98,99]. *Poria cocos*-derived polyporenic acid inhibits metastasis in PANC-1 by decreasing cell cycle protein 20 homolog (CDC20) [94]. A 60 µM dose of polyporenic acid inhibited significant PANC-1growth at 24 h [83,100]. Slingshot homologs (SSH)-derived cofilin dephosphorylation is linked by actin-depolymerization and these changes are accountable in invasive cancers [101]. Sennoside A acts as SSH inhibitors and helpful in treatments at metastasis conditions in PANC-1 and MIA PaCa-2 [101,102]. SH003 natural compounds (from *Astragalus, Angelica*, and *Trichosanthes*) showed VEGF-stimulated tumors angiogenesis. Its 10–50 µg/mL treatments significantly prohibited binding of VEGF to VEGFR2 at incubation of 24 h [103]. Also, DSGOST inhibits angiogenesis at 100 µg/mL, therefore these compounds (DSGOST, and SH003) cytotoxicity needs to be briefly evaluated for potential therapeutics [85,103,104]. 

Sugiol (diterpene) induces apoptosis by mitochondrial pathway and shows antiproliferative potential in Mia-PaCa2 via decline of Bcl-2 and rise of Bax expression [105]. In contrast, MMP expression down-regulated after 48 h at suigiol IC_50_ of 15 μM treatment that demonstrated maximum growth inhibition against Mia-PaCa2 pancreatic cancer cells [105,106]. A toosendanin inhibits migration and invasion of pancreatic cancer cells [107]. It increased E-cadherin (epithelial marker) and declines mesenchymal markers (Snail, vimentin, and ZEB). In addition, toosendanin diminished phosphorylation (AKT, mTOR, PRAS40, and p70S6K). Sennoside A at 10 µM dose produced maximum cytotoxicity for 20 m in pancreatic cancer cell Panc-1 [104,107]. Qingyihuaji (from *Amomi*, *Arisaematis, Hedyotdis*, *Gynostemmatis*, and *Scutellariae*) elevates lncRNA AB209630 expression and declines (EphB2, miR373, and Nanog) [108]. CFPAC-1 (gemcitabine-resistant) treated by qingyihuaji, and gemcitabine (30 ng/L) exhibited profound influence of cells migration and proliferations. 5,7-dihydroxy-3,6,8-trimethoxyflavone (flavone A) has been reported as plant-derived isomeric flavonoids, which induces apoptosis in highly tumorigenic cancers [109]. In addition, 40 μM flavone treatment for 9 h induced maximum apoptosis in MIA Paca-2 cell lines through intrinsic pathways, includes extracellular signal-regulated kinase (ERK), pS6, and Bcl-2 associated death promoter (BAD) proteins [109,110]. Hence, flavone A may be employed as a natural therapeutic molecule for treating pancreatic cancer. A synthesized novel benzo[4,5]imidazo[2,1-*b*]pyrimido[4,5-*d*][1,3]thiazine-2,4(3H)-diones and benzo[4,5]imidazo[2,1-*b*][1,3]thiazin-4-ones showed efficient anti-cancer activity towards A431, Panc1, and PC3 [111]. These compounds exhibited IC_50_ values of 21.6 and 31.9 µM against Panc1 compared to etoposide (25.2 µM). Overall, the details of the various therapeutic strategies, mechanisms, and potential applications are presented in Table 1.

### 3.7. Microbiome

The human microbiome is a diverse ecosystem of microorganisms that inhabit the human body. Association of microbes is linked by signal molecules’ communication [112,113,114]. The microbiome is estimated to consist of over 100 trillion microorganisms, the most common being bacteria, viruses, and fungi [115]. Recent studies advocate that the gut microbiome’s balance and heterogeneity may contribute to the advancement and development of diseases, including pancreatic cancer [116,117]. Therefore, this section confers how the microbiome impinges on pancreatic cancer progression.

Previous findings have suggested that the microbiome can influence many aspects of human health, including the progression and advancement of pancreatic cancer [113,118,119]. Studies have shown that pancreatic cancer had significant alternation in beneficial bacterial species, such as *Lactobacillus, Ruminococcus* and *Bifidobacterium,* compared to healthy individuals and may promote the development of cancer cells [120]. Recent studies have suggested that *Bifidobacterium* may have a protective effect against pancreatic cancer. Some studies have found that individuals with higher levels of *Bifidobacterium* in their gut microbiome have a lower risk of developing pancreatic cancer [121]. The proposed mechanism for this protective effect is that *Bifidobacterium* may help to maintain a healthy gut environment by promoting a balanced gut microbiome and reducing inflammation. This is important because chronic inflammation is a common risk aspect for advancing pancreatic cancer [122]. *Bifidobacterium* may also have direct anticancer effects by producing compounds that inhibit the growth of cancer cells. For example, some studies have found that *Bifidobacterium* can produce short-chain fatty acids (SCFAs), which have been shown to have anticancer properties [123]. *Lactobacillus* may also has a protective effect against pancreatic cancer [124]. These organisms produce lactic acid and hydrogen peroxide, which may possess anticancer properties[125]. Regarding the progression of pancreatic cancer, studies have shown that some bacteria may be engaged in the cancer spread and the production of substances that can make cancer cells worse. Gram-negative bacteria can produce substances called exotoxins that damage the pancreas and cause cancer cells to grow [126]. *Fusobacterium nucleatum*, a species of *Fusobacteria*, is present at higher levels in the pancreatic tumors of patients with pancreatic cancer than in healthy individuals [127]. The microbiome of tumors has been proposed to be linked to the advancement of cancer and the ability of chemotherapy to be effective [35].

Moreover, recent studies have shown that the microbiota may also be directly or indirectly responsible for developing drug resistance in pancreatic cancer [120]. The main factor is the ability of microorganisms to absorb and metabolize some anticancer drugs, rendering them ineffective [128]. While additional work is necessary to elucidate the exact mechanism involved, understanding the microbiome may also help to prevent, diagnose, and treat this deadly disease or human health [128,129,130]. Bacteria in the gut may also produce metabolites that promote the growth of cancer cells. Gut bacteria have been found to produce N-nitroso compounds [131], which are known to be carcinogenic [132]. In addition, the research provided evidence that *Helicobacter pylori* seropositivity and increased risk of pancreatic cancer [133]. Another hypothesis is that host-microbiome dysbiosis may also contribute to the development of pancreatic cancer [134,135]. Studies have shown that patients with pancreatic cancer often have a lower diversity of gut bacteria than healthy individuals [136]. The gut microbiome is a complex ecosystem, and its changes may have unintended consequences. Therefore, further research is necessary to recognize the role of the gut microbiome in understanding pancreatic cancer molecular pathways and how it can be modulated to prevent or treat the disease.

### 3.8. Clustered Regularly Interspaced Short Palindromic Repeats, and Associated Protein 9 (CRISPR/Cas9)

Significant progress has been made in developing new therapeutic modalities, including targeted, immune, and CRISPR/Cas9-guided therapy [137]. CRISPR/Cas9 technology is more efficient, less costly, and easier to implement than transcription activator-like effector nucleases (TALENs), mega-nucleases (MNs), and zinc-finger nucleases (ZFNs) [138]. It allows for the deletion, replacement, or addition of genomic sequences by inducing non-homologous end joining/homology-directed repair. CRISPR/Cas9 is routinely employed to silence genes in pancreatic cancer development. Using CRISPR/Cas9, Watanabe et al. [139], showed that KDM6A-lacking cells display hostile behavior in human PDAC cell lines. Reduced extracellular vesicle secretion and a less robust motile phenotype were examined by knockouts of ANXA1 in Mia PaCa2 [140,141,142]. Furthermore, the knocking down of GALNT3 using CRISPR/Cas9s in Capan1 cells resulted in fewer tumor spheres losing their self-renewal capacity and migrating less efficiently [141]. Furthermore, knocking down SphK1 in PAN02 cells using CRISPR led to higher proliferation and migration [143]. Moreover, CRISPR/Cas9 technology was utilized to knock down C1GALT1 in PDAC cells; these cells were more aggressive regarding proliferation, migration, tumorigenicity, metastasis, and the expression of Tn and sTn [144].

Loss-of-function (LoF) phenotypic screens using a small-guide RNA (sgRNA) library are a powerful method for discovering new protein roles under certain growth conditions [145]. Thousands of plasmids encoding sgRNAs targeting individual genes are used in one massive screening activity known as CRISPR pooled library screening. First, the sgRNAs must be transduced into enough cells using a lentivirus, or retrovirus. Ultimately, the transduced cells are cultivated in a controlled environment to achieve a desired phenotype. Cell’s DNA is extracted, and NGS is obtained after selection. The next step is for a computational tool like model-based analysis of genome-wide CRISPR-Cas9 knockouts to assess the NGS data by mapping it to a library containing the gene-specific gRNAs [145]. After transducing PANC1-cas9 to sgRNA library Brunello, several researchers have employed CRISPR assembled screening to target cancer [20]. The cells were then chosen with gemcitabine (100 nM) for six-days. Comparing the NGS data of selected samples than controls, PSMA6 was a differentially expressed gene. The subsequent research showed that PSMA6 suppression caused apoptosis and decreased spheroid development. Cells transduced with a TKO gRNA library and tested simultaneously revealed the fitness genes necessary for proliferation [20]. Using the BAGEL algorithm, each gene log Bayer factor (BF) was determined [146]. CTNNB1, LRP5, PORCN, TCF7L2, WLS, and the Frizzled (FZD) receptors FZD5, WNT7B, and WNT10A, were found to be necessary for HPAFII cell proliferation. This research was further confirmed in PaTU8988S and AsPC-1, establishing a Wnt pathway as a critical modulator for proliferations of cancer cells. In addition, the screening outcomes showed that Wnt—catenin signaling is specifically required in RNF43-mutant PDAC proliferations and/or survivals. Genome-wide CRISPR-Cas9 knockouts screens were also done in PATU8988T and PATU8902 cells to discover genes involved in proliferation/survival in MAPKi presence [147]. CRISPR/Cas9 gene-editing system has applications in both in vitro and in vivo settings. The ex-vivo therapy would involve isolating cells, modifying them in a lab, and reintroducing them to the patient. Injecting genetic materials directly into the body is possible with in vivo therapy [148]. In vivo, it was generated by injecting adenoviral-Cre or lentiviral-Cre vectors retrogradely into pancreatic ducts. This method can potentially pave the way for novel pancreatic cancer treatments [148]. Although multiple CRISPR/Cas9 clinical trials have been conducted, and the technology holds great promise for treating monogenic disorders, degenerative diseases, and HIV infection, there is still a long way to go before it is applied to treat pancreatic cancer patients. Many obstacles are yet to be pondered. For instance, which genes are the most significant ones to focus on? Furthermore, accuracy, efficiency, and safety improvements are necessary to fulfill the needs of clinical applications in pancreatic cancer.

## 4. Conclusions

Pancreatic cancer is primarily constrained by early diagnosis or potential therapy procedures. It is a potentially fatal disease that can spread to other body parts. If caught early, the disease can be treated with surgery and chemotherapy. However, some resist these treatments or do not respond to them at all. Fortunately, the cancer is responsive to targeted therapy agents that help it grow and invade healthy tissue. Targeted therapies for pancreatic cancer are a relatively new type of cancer treatment that targets specific molecules in the cancer cells. Important clinical progresses in diagnostic studies, surgical-techniques, and associated therapeutic strategies are sure to enhance survival of patients with pancreatic cancers. The better understanding of molecular manifestations involved in the pancreatic cancer and its progression can be helpful in developing promising therapeutic approaches.

## Figures and Tables

**Figure 1 biomedicines-11-01611-f001:**
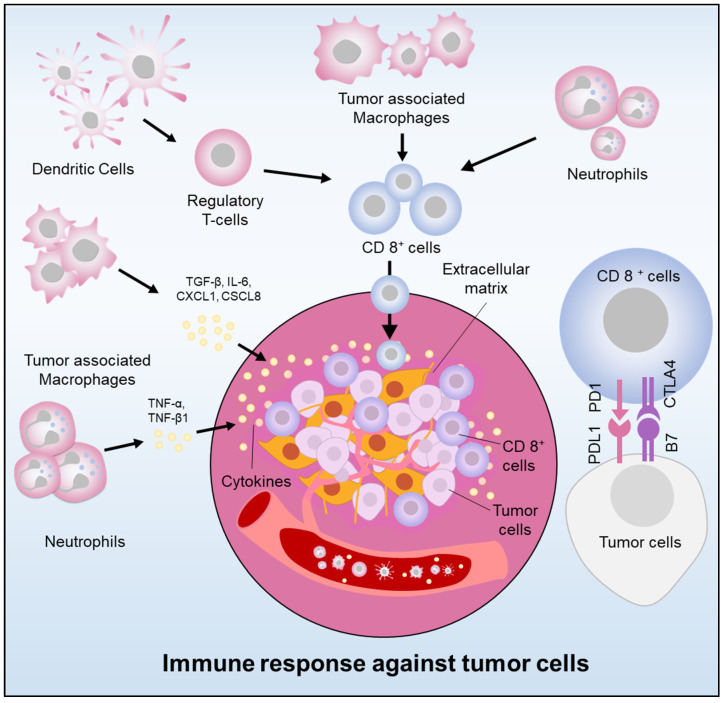
Immunoregulation in pancreatic cancer (Cancer cells elude the immune system by releasing cytokines and chemokines, recruiting immunosuppressive cells, and expressing PDL-1. Immune cell infiltration also stimulates tumor growth).

**Figure 2 biomedicines-11-01611-f002:**
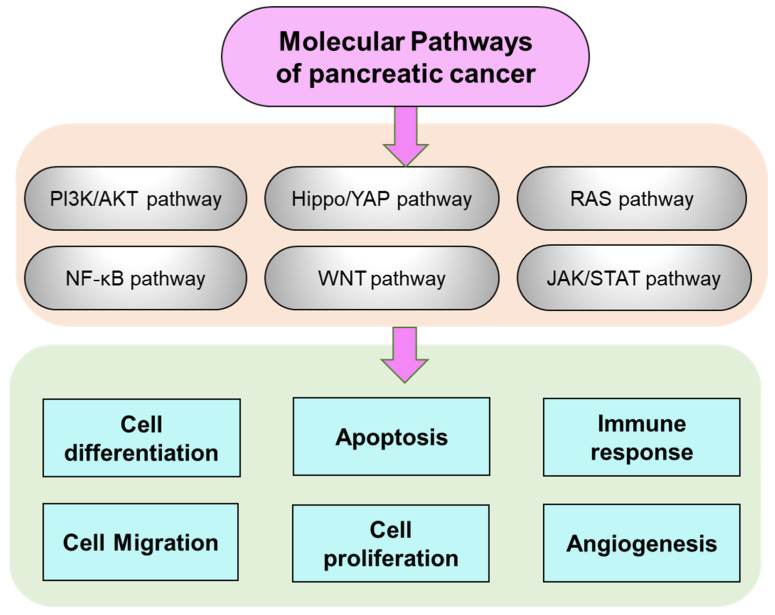
Molecular pathways of pancreatic cancer.

**Table 1 biomedicines-11-01611-t001:** Various therapeutic approaches for pancreatic cancer treatments.

Molecules/Biomolecules	Mechanism	Remarks/Application	Reference
miR-203	DUSP5 downregulating expression	Proliferation, migration, and colony-forming potential of PANC-1 inhibition	[60]
Piperazine-tethered phthalazines	Selective CDK1 inhibitors	Potential activity against pancreatic cancer adenocarcinoma: MDA-PATC53 (IC_50_ = 0.51–0.88 μM), and PL45 (IC_50_ = 0.74–1.14 μM), and CDK1 inhibitory activity with IC_50_ of 36.8–44.5 nM	[24]
1-Piperazinylphthalazines	VEGFR-2 inhibitors	IC_50_ of 0.30–0.40 μM for compounds 16 k and 21 d towards VEGFR-2	[63]
Oncolytic adenovirus OBP-702	p53 overexpression	Induces ICD and antitumor immune responses in PDAC with distinct p53 status	[64]
Chemically modified MIR143-3p	Suppressing the entire RAS network	MIR143#12 IC_50_ values of 63.25 and 4.74 nM at 72 and 96 h of incubation against Panc-1, respectively	[65]
Carrier-free prodrug nanoparticles (F68-FDOX)	Cytotoxicity against cancer cells	High drug loading above 50%, and showed a broad therapeutic spectrum against colon, breast, and pancreatic cancers	66]
Dextran-coated maghemite nanoparticles	Alteration in heat shock proteins (HSPs) and p53 protein expression	Nanoparticles (56 μg/mL) exposure reduced 50% of PANC-1 cell viability after 72 h	[67]
Lipid nanoparticles (LNP_DTX-P_) on gold nanoparticles	Enhanced uptake of LNP_DTX-P_ by pancreatic cancer cells, and exhibtes synergetic radiosensitization effects	2-fold enhancement in nanoparticles uptake by LNP_DTX-P_-treated tumour cells compared to pure nanoparticles	[68]
Magnetic nanoparticles and hyperthermia	Increase DNA double strand breaks by radiosensitization effects and ROS production	The combined treatment of magnetic nanoparticles and hyperthermia enhanced cell death at 6 h in BxPC3 pancreatic tumor cells compared to irradiation or nanoparticles adminstration	[69]
Polyethyleneglycol-betulinic acid (PEG-BA) polymer-drug conjugate	Induces apoptosis mediated death of MIA-PaCa-2 by arresting sub-G1phase of cell cycl and involves in anti-inflammatory and antioxidant activities	PEG-BA (4 μM) treated cells showed over-expression of the proapoptotic genes *TNF* (23.7-fold) and *CASPASE 3* (12,060-fold), and exhibits IC_50_ of 15.6 μM compared to BA-only of >100 μM for antioxidant potential	[70]
S-1 adjuvant chemotherapy	Total dose intensity-derived survival prediction	Maintenance of dose intensity >60% in S-1 adjuvant chemotherapy improves survival of pancreatic cancer patients	[71]
Protein aggregate magnesium-ammonium phospholinoleate-palmitoleate anhydride	Suppression of abnormal cell proliferation altering TLR4 signaling pathway	P-MAPA-based showed histopathological repair in 40% of rats, and P-MAPA/gemcitabine-associated treatment was effective in reducing neoplastic lesion progression, and enabling histopathological improvement in 80% of rats	[19]
LSKL peptide	Thrombospondin-1 (TSP-1) inhibitors	TSP-1 promotes Smad4 expression deficiency, and malignant potential by activation of TGF-β signal in PDAC	[72]
Curcumin	Sensitization pf cancer cells to gemcitabine by assuaging expressions of PRC2 subunit EZH2, and lncRNA PVT1 and overcoming drug-resisance	IC_50_ value 8 and 20 µM for curcumin against BxPC3 and Panc1 cells, respectively	[74,80,81]
Chaetospirolactone	Induction of apoptosis by upregulating apoptotic proteins such as c-caspases (3, 8, and 9), and downregulating *EZH2* gene	100 nM CSL treatment for 24 h reverses TRAIL resistance in PANC-1 cells via epigenetic regulation of DR4	[80,83]
Cordifoliketones A	Apoptosis induction through upregulating Bad, Bax, and caspases (3, 8, and 9), and downregulating oncogenes (Bcl-2, and Bcl-xL)	Minimum IC_50_ of 4.18 μg/mL of cordifoliketones A is required for the maximum inhibition of cell growth in PANC-1 AsPC-1, BxPC-3 and PANC-1 after 48 h of treatment	[83,84]
Danggui-Sayuk-GaOsuyu-SaenggangTang (DSGOST)	Inhibition of migration, and tube formation by upregulating caspage-3 and downregulating p-IKKα/β, p-IκBα, p-NF-κB, p-AKT, p-VEGFR2, p-FAK, p-SRC, and MMP-9	DSGOST at dose-100 µg/mL inhibited maximum cell growth after 2 h of tratement in PANC-1	[83,85]
Dicatenarin	Induces apoptosis by inducing reactive oxygen species and increased induction of caspase-3	IC_50_ values of 12 µg/mL against MIA PaCa-2 cell line	[86,87]
Diphyllin derivatives (amino derivative 15)	Cell cycle arrests at G0/G1 phase	69-fold more potent than diphyllin with IC_50_ of 3 nM against pancreatic cancer CFPAC-1 cells	[88]
Diosgenin	Inhibition of tumor growth by upregulating tumor suppressor PTEN and downregulating oncogene EZH2, and vimentin using Patu8988 and Panc-1 cell lies	75 μg/mL diosgenis for 72 h suppreses upto 70% growth of Patu8988 and Panc-1 cell lies	[89,90]
Echinacoside	Induction of apoptosis by upregulating ROS, Bax, p38 and downregulating MMP, JNK, and ERK1/2 in SW1990 cell lines	Maximum growth inhibition was observed at 72 h using 100 µM of echinacoside for SW1990 cell lines	[83,91]
Elemene	Cell proliferations inhibition, and cell cycle arrests by upregulating tumor suppressor gene p53 and down regulating oncogene Bcl-2	60 µg/mL od elemene induces more than 85% of cell death after 72 h of tratment in BxPC-3 and Panc-1 cell lines	[92,93]
Methyl4-(2-isothiocyanatoethyl)benzoate	Apoptosis induction by upregulating ROS and downregulating oncogene glutathione (GSH)	Maximum appoptosis at dose 10 µM methyl4-(2-isothiocyanatoethyl)benzoate after 72 h treatment in Panc1 and Capan2 cell lines	[94,95]
Monogalactosyl diacylglycerol (MGDG)	Apoptosis induction by upregulating cytochrome c, c-PARP, Bax, and c-caspase-3, and downregulating Bcl-2	Maximum cell death was observed at IC_50_ of 25.6 in PANC-1, BxPC-3, MIAPaCa-2 and AsPC-1 cells for 72 h treatment	[83,96]
Piperlongumine	Apoptosis induction by upregulating procaspase-3, c-PARP, and downregulating Bcl-2, Bcl-xL, survivin, XIAP, miR-27a, and miR-17/miR-20a	Piperlongumine at dose 40 µmol/L inhibit maximum cell growth in PANC-1 after 72 h of treatment	[97,98,99]
Polyporenic acid	Inhibition of metastasis by down regulating of oncogene CDC20	Polyporenic acid at dose 60 µM, for 24 h has shwon mximum inhibiot of cell growth in PANC-1	[83,100]
Sennoside A	Cells invasion and migration inhibition through downregulating p-cofilin	Sennoside A at 10 µM dose produced maximimum cytotoxicity for 20 m in pancreatic cancer cell Panc-1	[101,102]
SH003	Angiogenesis inhibition by upregulating c-caspase-3, and downregulating p-VEGFR2, MMP-9, p-FAK, p-SRC, p-ERK, p-AKT, and p-STAT3	SH003 at dose 20 µg/mL has shown maximimun cytotoxic effects for 62 h of treatment in PANC-1	[103,104]
Sugiol	Apoptosis induction, cell cycle arrest, and increase of ROS production by upregulating Bax and downregulating Bcl-2, and MMP	At IC_50_ of 15 μM of suigiol treatment for 48 h, suppres maximum growth of Mia-PaCa2 pancreatic cancer cells	[105,106]
Toosendanin	Cells invasion and migration inhibition by upregulating E-cadherin and downregulating Vimentin, ZEB1, Snail, p-AKT, p-PRAS40, p-mTOR, and p-p70S6K	Toosendanin at concentration of 200 nM, decreased 7–8 fold in the migratory capacity of PANC-1 and AsP pancreatic cancer cells, after 5 days of treatment	[104,107]
Qingyihuaji	Cells invasion and migration inhibition by upregulating lncRNA AB209630 and downregulating miR-373, EphB2, and Nanog	Qingyihuaji at 40 µg/L demonstarted maximum inhibition of cell growth for 72 h in CFPAC-1 cell lines	[83,108]
5,7-dihydroxy-3,6,8-trimethoxyflavone (flavone A)	Apoptosis induction and cell cycle arrests by upregulating p-ERK, p-c-JUN and downregulating pS6, p-Bad, Bcl-xL, and Bcl-2	40 μM flavone treatment for 9 h iduced maximum appoptosis in pancreatic cancer MIA Paca-2 cell lines.	[109,110]
2,2-dimethyl-5-(4-nitrophenyl)-1,2,3,12a-tetrahydro4H-benzo[d]benzo[4,5]imidazo[2,1-b][1,3]thiazine-4-one	Cytotoxicity against cancer cells	IC_50_ values of 21.59 µM against Panc1 compared to etoposide (25.19 µM)	[111]
1,3-dimethyl-5-(p-tolyl)-1,12a-dihydro-2H-benzo[4,5]imidazo[2,1-b]pyrimido[4,5-d][1,3]thiazine-2,4(3H)-dione	Cytotoxicity against cancer cells	IC_50_ values of 31.87 µM against Panc1 compared to etoposide (25.19 µM)	

## Data Availability

Not applicable.

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
