# Peer review of "Therapeutic Approaches in Pancreatic Cancer: Recent Updates"

_biomedicines, 2023, doi:10.3390/biomedicines11061611_

Round 1
Reviewer 1 Report
The authors propose a proper literature review; there are some points to review:
- miRNAs: synthetic antagomirs or miRNAs could at least be cited as hypotheses
- the paragraph on the microbiota seems very speculative to me; if maintained I would consider it as a hypothesis, I would eliminate what concerns prevention which is not the object of the work
- I would suggest considering a paragraph on the ketogenic diet which is increasingly being used in support of other anticancer therapies as well
It needs slight revisions.
Author Response
Reviewer #1:
The authors propose a proper literature review; there are some points to review:
- miRNAs: synthetic antagomirs or miRNAs could at least be cited as hypotheses
Response: Thank you for your feedback on the literature review, including synthetic antagomirs or miRNAs cited as hypotheses is indeed a valuable suggestion. Therefore, we have incorporated these points in this review to provide a more comprehensive analysis of the topic as follows:
(lines, 176-190) “Micro-RNAs (miRNAs) or synthetic antagomirs are ncRNAs composed of ~22 long nucleotides. These are proposed to be associated with regulatory functions, i.e., cell proliferation, apoptosis, and autophagy. In addition, studies have shown that miRNAs such as miR-203 possess antitumor potential by regulating gene expressions [36]. However, miRNAs released in the bloodstream can be used as markers or indicators to monitor pancreatic cancer progression or aggressiveness. The role of a few miRNAs in cancer-transforming cell metabolism includes as follows: (i) miR17-92 and miR-21 in proliferation inhibition, (ii) miR-126 in anti-oncogene action inhibition, (iii) miR-15b, and miR-155 in mutations accumulation, (iv) let-7d, miR-23b, miR-126, and miR-200c in inflammation promotion, (v) miR-10b and miR-29 in metastasis activation, (vi) miR-21 and miR17-92 in the immune cells elimination inhibition, and (vii) let-7, miR-16, miR-21, and miR-221/222 in replication immortality [36]. miR-203 helps suppress cell invasion and migration by caveolin-1 and is down-regulated in pancreatic cancers [4]. Various miRNAs involved in pancreatic cancer are miR-21, miR-155, miR-221, miR-222, miR-376a, and miR-301[36].”
- Smolarz, B.; Durczyński, A.; Romanowicz, H.; Hogendorf, P. The role of microRNA in pancreatic cancer. Biomedicines 2021, 9, 1322.
- the paragraph on the microbiota seems very speculative to me; if maintained I would consider it as a hypothesis, I would eliminate what concerns prevention which is not the object of the work
Response: Thank you for your valuable feedback on the paragraph discussing the microbiota in our literature review. We acknowledge your concern regarding the speculative nature of the content, and we agree that it would be more appropriate to present it as a hypothesis rather than stating it as established facts. Furthermore, we have carefully considered your suggestion to eliminate the sections related to prevention from our review as follows:
(lines, 466-467) “Association of microbes is linked by signal molecules' communication [112-114].”
(lines, 486-488) “Lactobacillus may also has a protective effect against pancreatic cancer [124]. These organisms produce lactic acid and hydrogen peroxide, which may possess anticancer properties[125]”
(lines, 494-497) “The microbiome of tumors has been proposed to be linked to the advancement of cancer and the ability of chemotherapy to be effective [35].
Moreover, recent studies have shown that the microbiota may also be directly or indirectly responsible for developing drug resistance in pancreatic cancer [120].”
(lines, 499-501) “While additional work is necessary to elucidate the exact mechanism involved, understanding the microbiome may also help to prevent, diagnose, and treat this deadly disease or human health [128-130].”
Zhang, X.; Liu, Q.; Liao, Q., Zhao, Y. Pancreatic Cancer, Gut Microbiota, and Therapeutic Efficacy. J. Cancer 2020, 11, 2749–2758.
- I would suggest considering a paragraph on the ketogenic diet which is increasingly being used in support of other anticancer therapies as well
Response: Thank you kindly for your suggestion regarding the inclusion of a paragraph on the ketogenic diet in our study. We truly appreciate your input and the valuable insights you have shared. However, after careful consideration, we have decided to focus our review on other aspects, and therefore, we kindly request that you allow us to omit the section from the manuscript. We sincerely appreciate your understanding and support.

Reviewer 2 Report
Therapeutic approaches in the pancreatic cancer were reviewed with recent updates. This review describes the recent development in pancreatic cancer progression, mechanisms, and therapeutic approaches, including molecular techniques and biomedicines for effectively treating cancer. The major structure of the manuscript is composed of 2. Pancreatic cancer molecular manifestation and pathways regulation and 3. Therapeutic strategies (3.1. Non-coding RNAs, 3.2. Cyclin-dependent kinases, 3.3. Oncolytic virus, 3.4. Nanomedicine, 3.5. Adjuvants, Immunological Targets and Peptide, 3.6. Natural Bioactive and Organic Compounds, 3.7. Microbiome and 3.8. Clustered Regularly Interspaced Short Palindromic Repeats, and Associated Protein 9 497 (CRISPR/Cas9)).
The manuscript is interesting but it needs major revision. First of all novelty of the review should be pointed out at the end of introduction part with respect to previously published reviews on this topic. In addition, updates should be defined by the range of years that are covered (from the title to the introduction part and in overall manuscript).
Does any other review on Pancreatic cancer exists? Just a short view even on Google scholar reveals the review papers on this topic such as: https://doi.org/10.3892/ol.2017.5572; Signaling pathways in pancreatic cancer. - Abstract - Europe PMC; https://doi.org/10.3389/fonc.2012.00006, others. The authors should mention these and other reported reviews in the introduction part. They also need to point out the novelty of present review in comparison to the already published reviews.
Minor English language corrections would improve the manuscript.
Author Response
Manuscript: biomedicines-2387092
Title : Therapeutic Approaches in the Pancreatic Cancer: Recent Up-dates
Reviewer #2:
Therapeutic approaches in the pancreatic cancer were reviewed with recent updates. This review describes the recent development in pancreatic cancer progression, mechanisms, and therapeutic approaches, including molecular techniques and biomedicines for effectively treating cancer. The major structure of the manuscript is composed of 2. Pancreatic cancer molecular manifestation and pathways regulation and 3. Therapeutic strategies (3.1. Non-coding RNAs, 3.2. Cyclin-dependent kinases, 3.3. Oncolytic virus, 3.4. Nanomedicine, 3.5. Adjuvants, Immunological Targets and Peptide, 3.6. Natural Bioactive and Organic Compounds, 3.7. Microbiome and 3.8. Clustered Regularly Interspaced Short Palindromic Repeats, and Associated Protein 9 497 (CRISPR/Cas9)).
Response: Thank you for your insightful comment regarding our review on therapeutic approaches in pancreatic cancer. We appreciate your recognition of the recent updates and comprehensive coverage of the topic.
The manuscript is interesting but it needs major revision. First of all novelty of the review should be pointed out at the end of introduction part with respect to previously published reviews on this topic. In addition, updates should be defined by the range of years that are covered (from the title to the introduction part and in overall manuscript).
Response: We sincerely appreciate your suggestions for improvement and assure you that we will address the points you raised. We have incorporated the novelty of the review at the conclusion of the introduction section as follows:
(lines, 75-83) “Although plenty of excellent reviews are available on pancreatic cancer [32-37], this review stands out as it offers an in-depth analysis of the most recent studies and advancements in the field. It thoroughly examines the current understanding of the progression of this disease and explores a wide range of therapeutic strategies that show potential in treating pancreatic cancer. These strategies include traditional chemotherapy, surgical interventions, and cutting-edge approaches such as immunotherapy and targeted therapies. This review evaluates these treatment option’s effectiveness, limitations, and possible side effects. It aims to provide a comprehensive perspective on the advancements made in pancreatic cancer treatment.”
Does any other review on Pancreatic cancer exists? Just a short view even on Google scholar reveals the review papers on this topic such as: https://doi.org/10.3892/ol.2017.5572; Signaling pathways in pancreatic cancer. - Abstract - Europe PMC; https://doi.org/10.3389/fonc.2012.00006, others. The authors should mention these and other reported reviews in the introduction part. They also need to point out the novelty of present review in comparison to the already published reviews.
Response: We apologize for any oversight in not mentioning these existing reviews in our manuscript. Based on your feedback, we have revised our introduction paragraph to include a specific paragraph highlighting the significance of our review in comparison to the previously published manuscripts. By doing so, we aim to provide readers with a clear understanding of the unique contributions and advancements presented in our work. We thank you for your valuable input. The information has been added in the last paragraph of the manuscript introduction section as follows:
(lines, 75-83) “Although plenty of excellent reviews are available on pancreatic cancer [32-37], this review stands out as it offers an in-depth analysis of the most recent studies and advancements in the field. It thoroughly examines the current understanding of the progression of this disease and explores a wide range of therapeutic strategies that show potential in treating pancreatic cancer. These strategies include traditional chemotherapy, surgical interventions, and cutting-edge approaches such as immunotherapy and targeted therapies. This review evaluates these treatment option’s effectiveness, limitations, and possible side effects. It aims to provide a comprehensive perspective on the advancements made in pancreatic cancer treatment.”
- Reddy, S.A. Signaling pathways in pancreatic cancer. Cancer J. 2001, 7, 274–286.
- Iovanna, J.; Mallmann, M.C.; Goncalves, A.; Turrini, O.; Dagorn, J.-C. Current knowledge on pancreatic cancer. Front. Oncol. 2012, 2, 6.
- Gong, J.; Sachdev, E.; Robbins, L.A.; Lin, E.; Hendifar, A.E.; Mita, M.M. Statins and pancreatic cancer. Oncol. Lett. 2017, 71, 1035–1040.
- Zhang, X.; Liu, Q.; Liao, Q., Zhao, Y. Pancreatic cancer, gut microbiota, and therapeutic efficacy. J. Cancer 2020, 11, 2749–2758.
- Smolarz, B.; Durczyński, A.; Romanowicz, H.; Hogendorf, P. The role of microRNA in pancreatic cancer. Biomedicines 2021, 9, 1322.
- Kolbeinsson, H.M.; Chandana, S.; Wright, G.P.; Chung, M. Pancreatic cancer: A review of current treatment and novel therapies. J. Invest. Surg. 2023, 36, 2129884.

Reviewer 3 Report
This is a well-written article based on the latest literature in the field, however, I have reservations about the figures which convey very little information. It is unclear what the arrows on Figure 1 represent and what the relationship is between the cells depicted in the figure. Additionally, Figure 2 is completely incomprehensible. There is a lack of citation at line 162.
I am not a native speaker, so I do not notice subtle language errors, but for me, the article is written comprehensibly from a linguistic point of view.
Author Response
Manuscript: biomedicines-2387092
Title : Therapeutic Approaches in the Pancreatic Cancer: Recent Up-dates
Reviewer #3:
This is a well-written article based on the latest literature in the field, however, I have reservations about the figures which convey very little information. It is unclear what the arrows on Figure 1 represent and what the relationship is between the cells depicted in the figure. Additionally, Figure 2 is completely incomprehensible. There is a lack of citation at line 162.
Response: Thank you for your feedback on our article. We appreciate your positive comments on the overall quality of the manuscript. We also thank you for bringing some concerns to our attention.
- We apologize for the lack of clarity in Figure 1. We understand that the arrows and the relationship between the depicted cells need to be better defined. The arrow in the figure explains the activation of the cells or release of components from the cells. We have updated the figure for better clarity as follows:
Figure.1: Immunoregulation in pancreatic cancer (Cancer cells elude the immune system by releasing cytokines and chemokines, recruiting immunosuppressive cells, and expressing PDL-1. Immune cell infiltration also stimulates tumor growth).
- Regarding Figure 2, we acknowledge that it is currently incomprehensible. We apologize for this confusion, and we made necessary revisions to the image to ensure that the figure becomes clear and easily interpretable as follows:
Figure 2. Molecular pathways of pancreatic cancer.
- Furthermore, we apologize for the missing citation at line 162. We understand the importance of proper referencing and will promptly address this issue by including the appropriate citation at the specified line as follows:
(line, 170) “~ [60],”

Round 2
Reviewer 2 Report
The authors revised the manuscript according to the comments.
-